# A Novel Modulator of Resistance for Oxaliplatin-Based Therapy for Colorectal Cancer: The ESCRT Family Member VPS4A

**DOI:** 10.3390/cells14120929

**Published:** 2025-06-19

**Authors:** Noha M. Abdelrazik, Anjana Patel, Andrew Conn, Christopher W. Sutton, Sriharsha Kantamneni, Steven D. Shnyder

**Affiliations:** 1Institute of Cancer Therapeutics, University of Bradford, Bradford BD7 1DP, UK or noha.magdy@fop.usc.edu.eg (N.M.A.); a.patel62@bradford.ac.uk (A.P.); c.w.sutton@bradford.ac.uk (C.W.S.); S.Kantamneni@bradford.ac.uk (S.K.); 2Faculty of Pharmacy, University of Sadat City, El Sadat City 32897, Egypt; 3Department of Medical Oncology, Bradford Royal Infirmary, Bradford Teaching Hospitals NHS Foundation Trust, Bradford BD9 6RJ, UK; andrew.conn@bthft.nhs.uk

**Keywords:** colorectal cancer, drug resistance, oxaliplatin, endosomal sorting complex required for transport (ESCRT) protein family, vacuolar protein sorting-associated protein A4

## Abstract

Drug resistance is still one of the main challenges for the treatment of colorectal cancer (CRC). Whilst some resistance mechanisms are well known, from the static therapy success rate, clearly, still much is undiscovered. Intracellular transport mechanisms have attracted attention as having a possible role in drug resistance, and here, the Endosomal Sorting Complex Required for Transport (ESCRT) protein family is studied as a source of drug resistance modulation using human CRC cell lines and clinical material. From an initial screening of ESCRT proteins in a panel of 10 CRC wild-type cell lines using immunoblotting, Vacuolar Protein Sorting-Associated Protein A4 (VPS4A) was identified as being consistently highly expressed, and it was selected for further investigation. Immunohistopathological evaluation in a small panel of CRC patient samples demonstrated high expression in the tumor epithelium compared to normal intestinal epithelium. The knockdown of VPS4A resulted in enhanced sensitivity of cells to oxaliplatin, and it was subsequently seen that oxaliplatin-resistant sublines had significantly higher VPS4A expression than their wild-type variants. In addition, it was demonstrated that a small molecule inhibitor of VPS4A, aloperine, could interact synergistically with oxaliplatin to enhance its sensitivity in an oxaliplatin-resistant cell line. We hypothesize from initial RNA sequencing analysis that the mechanism of action of VPS4A modulation is through depleting levels of the drug efflux transporter MRP2 in the cell, preventing oxaliplatin egress and increasing cell exposure to the drug. The evidence presented here thus indicates that ESCRT machinery, specifically VPS4A, may act as a modulator of oxaliplatin resistance in CRC.

## 1. Introduction

For patients with colorectal cancer (CRC), whilst there has been an improvement in the relative 5-year survival rate from 50% during the mid-1970s to around 65% today, over the past 25 years, this rate has remained static. A key reason for this is acquired drug resistance to the agents used in the two most common CRC treatment regimens, FOLFOX (5-fluorouracil (5-FU), oxaliplatin (Oxa) and leucovorin) and FOLFIRI (5-FU, irinotecan (Iri) and leucovorin).

Although some of the mechanisms of resistance to the drugs used in CRC treatment are well known, for example, increased thymidylate synthetase expression in the case of 5-FU, reduced topoisomerase expression for Iri, or decreased expression of transporters belonging to the SLC22 family for Oxa [1,2,3], there is still much that is undiscovered. Intracellular transport mechanisms have come under attention as having a possible role in drug resistance in cancer such as copper transport systems for platinum compounds [4], or transient receptor channel proteins [5]. One such mechanism which has had attention is the Endosomal Sorting Complex Required for Transport (ESCRT) protein family [6,7].

ESCRTs are a family of proteins that are involved in multiple physiological and biological processes such as cell division and degradation of membrane proteins, membrane repair, and viral budding. One of the main functions of ESCRTs is to sort ubiquitinated proteins into late endosomes intralumenal vesicles to form multivesicular bodies (MVBs), and later fuse with lysosomes, targeting the degradation of these proteins. ESCRT complex proteins mediate this primary sorting event and regulate the stability of receptors at the surface of cells [8]. Besides their physiological functions, many diseases have been associated with misregulation and mutations in ESCRT machinery, including cancer, infection, neurodegenerative diseases, and wound healing [9]. The ESCRT family consists of four different complexes of proteins: ESCRT-0 (HRS and STAM-1), ESCRT-I (TSG101), ESCRT-II, and ESCRT–III (CHMP’s), as well as accessory proteins such as Alix and the Vacuolar Protein Sorting-Associated Proteins VPS4A and VPS4B. VPS4 proteins are ATPases that mediate the final steps of membrane fission and protein sorting as part of the ESCRT machinery. They disassemble ESCRT-III filaments, which are vital for forming MVBs and the release of intraluminal vesicles, ultimately leading to the sorting and degradation of various cellular proteins, including those involved in cancer development and progression [10]. 

In terms of cancer, whilst there are multiple studies which have covered the involvement of ESCRT machinery in biological processes, their role in cancer is less extensively investigated, and the underlying molecular mechanisms have been elucidated for only a few cases [11,12,13]. Large-scale screening for cancer vulnerabilities within the Sanger’s Project Score [14] and the DRIVE projects [15] showed that particular cancer cell lines had greater sensitivity when VPS4A expression was dysregulated [16]. Some studies regarding ESCRT-III members have demonstrated that CHMP2A sensitized glioblastoma stem cells to death mediated by natural killer cells [17]. Additionally, CHMP2A inhibition led to the activation of caspase 8-induced apoptosis in osteosarcoma and neuroblastoma cells [18]. However, so far, the link between the specific ESCRT family members and resistance to chemotherapy has not been extensively investigated.

Therefore, in this study, the potential of ESCRTs as modulators of drug resistance was explored, identifying a family member consistently overexpressed in a human CRC cell line panel, VPS4A. Expression in clinical samples and the effect of VPS4A knockdown and inhibition on response to drugs routinely used to treat CRC in the clinic were then explored.

## 2. Materials and Methods

### 2.1. Materials

A panel of ten CRC cell lines (COLO 205, DLD-1, HCC2998, HCT116, HT29, HT55, KM12, LS174T, SW480, and SW620), plus a normal intestinal epithelial cell line, HIEC-6, were used in these studies. DLD-1 and HIEC-6 were obtained from ATCC (LGC Standards, Teddington, UK), HT55, SW480, and LS174T from ECACC (Salisbury, UK), and COLO 205, HT29, HCC2998, HCT116, KM12, and SW620 from the National Cancer Institute Department of Cancer Treatment and Diagnosis Tumour Repository (Frederick, MD, USA).

The cells were all maintained in RPMI 1640 culture medium supplemented with 10% (*v*/*v*) fetal bovine serum (FBS), 1 mmol/L sodium pyruvate, 2 mmol/L of L-glutamine (all from Merck, Gillingham, UK), apart from HIEC-6, which was grown in Opti-Mem I reduced serum Medium supplemented with 20 mM HEPES, 10 mM GlutaMax, 10 ng/mL Epidermal growth factor (EGF), and 4% fetal bovine serum (FBS) (all from Thermofisher Scientific, Loughborough, UK). All cell lines were incubated at 37 °C in 5% CO_2_. Phosphate-buffered saline (PBS; Severn Biotech, Kidderminster, UK) was used for washing steps.

Oxaliplatin (Oxa), 5-Fluorouracil (5-FU), Irinotecan (Iri), and Aloperine (Alo) were all purchased from Selleckchem (Waltham Abbey, UK). Oxa, 5-FU, and Alo were initially prepared as a 17 mM, 50 mM, and 27 mM stock solutions in sterile distilled water, respectively, whilst Iri was initially prepared as a 21 mM stock in DMSO. In all cases, aliquots were stored at −20 °C until use. 

The following primary antibodies were used for immunoblotting and immunohistochemistry in this study: rabbit polyclonal anti-VPS4 A/B antibody (#SAB4200025, Merck); rabbit polyclonal anti-VPS4A antibody (#14272-AP, Proteintech, Manchester, UK); rabbit polyclonal anti-CHMP6 antibody (#ab76929, abcam, Cambridge, UK); recombinant rabbit monoclonal anti-TSG101 (#ab125011, abcam); rabbit polyclonal anti-CHMP2B antibody (#ab33174, abcam); rabbit polyclonal anti-STAM2 antibody (#HPA035528, Atlas antibodies, Stockholm, Sweden); rabbit polyclonal anti-VPS36 antibody (#HPA043947, Atlas antibodies); rabbit polyclonal anti-VPS37A antibody (#PA5–55160, Thermofisher Scientific); mouse monoclonal anti-VPS25 antibody (Santa Cruz, Middlesex, UK); rabbit polyclonal anti-ALIX antibody (#12422–1-AP, Proteintech); recombinant rabbit monoclonal anto-MRP2 antobidy (#ab172630, abcam); mouse monoclonal anti-GAPDH antibody (#6004–1-Ig, Proteintech), and rabbit polyclonal anti-GAPDH antibody (#10494–1-AP, Proteintech). All antibodies were aliquoted and stored in −20 °C until use.

### 2.2. Immunoblotting

Cells were lysed with 1 X RIPA buffer (Merck), supplemented with cOmplete™, EDTA-free Protease inhibitor cocktail (Merck). Protein concentration was measured using a BCA Kit (Thermofisher Scientific). Protein expression analysis was carried out by immunoblotting. A sample of whole cell lysate from each cell line was separated by SDS-PAGE electrophoresis in 10% acrylamide gels using a NuPAGE 4x LDS sample buffer (Thermofisher Scientific), followed by electroblotting onto PVDF blotting membrane (0.45 μm) (Merck) using constant voltage 100 V for 2 h. Membranes were blocked in 5% bovine serum albumin (BSA) (VWR, Lutterworth, UK). Rabbit polyclonal anti-VPS4 A/B antibody (#SAB4200025, Merck) at 1:1000 dilution and mouse monoclonal anti-GAPDH antibody (Proteintech) at 1:5000 dilution were used for protein expression. IRDye^®^ 800CW goat anti-rabbit IgG and IRDye^®^ 680RD donkey anti-mouse IgG secondary antibodies were used at 1:5000 dilution (LI-COR Biosciences, Cambridge, UK). Finally, the membrane was visualized, and images were taken using Odyssey^®^ Imagers (LI-COR Biosciences). Densitometric analysis was performed using Empiria Studio^®^ Software (LI-COR Biosciences).

### 2.3. Clinical Material

The study protocol was approved by the University of Bradford’s Independent Scientific Advisory Committee (reference: application/21/107). Ethical approval was given by Leeds (East) Research Ethics Committee, UK, reference: 22/YH/0111.

Archival formalin-fixed, paraffin-embedded (FFPE) diagnostic samples from colorectal cancer patients taken prior to the commencement of palliative chemotherapy were provided by Bradford Teaching Hospitals NHS Trust. The clinical details for the patients included in the study are given in Table 1.

On receipt of the archival FFPE blocks, 5 µm thick sections were taken using a Leica microtome and collected onto 3-Aminoproyl triethoxysilane (APES; Merck)-coated microscope slides. After drying, slides were stored in a dust-free environment at room temperature until use.

### 2.4. Immunohistochemistry

The immunohistochemistry protocol was optimized such that the same steps were followed for each biomarker, with the only difference being the concentration of the primary biomarker antibody. Unless stated, all steps were carried out at room temperature. Samples were initially dewaxed and rehydrated using a series of xylenes and ethanols (Thermofisher Scientific) through to distilled water. Endogenous peroxidase was then quenched in 1% hydrogen peroxide (Merck) in distilled water for 30 min followed by two washes in PBS. Antigen retrieval was then carried out by immersing the slides in sodium citrate buffer, pH 6.0, and microwaving at a medium setting for 15 min and then allowing them to cool for 30 min before proceeding to the next step. After 2 further washes in PBS, blocking with 1.5% normal goat serum in PBS (NGS; abcam) was carried out for 20 min, followed by administration of the optimized concentration of the specific primary antibody, polyclonal anti-VPS4A (Proteintech) diluted 1:100 in the blocking serum. Sections were then incubated overnight in a humidified chamber at 4⁰C. The following day, samples were washed twice in PBS followed by incubation with biotinylated goat anti-rabbit IgG secondary antibody (BP-9100–50, Vector Laboratories, Peterborough, UK), diluted 1:100 in PBS for 30 min. After two subsequent washes in PBS, 30 min incubation in avidin-biotin complex reagent (Vector Laboratories), and a further couple of PBS washes, a DAB staining kit (Vector Laboratories) was applied for 3 to 5 min. Slides were then washed in running tap water, Harris’s Haematoxylin counterstain (Merck) applied for 20 s, followed by further washing in tap water, immersion in Scott’s Tap Water (Merck) for 2 min, and a further tap water wash. Slides were then dehydrated and mounted as described in the previous section.

### 2.5. Microscopy, Image Capture, and Analysis

Sections were viewed using a Leica DMLS Microscope with images captured using a Leica DFC295 digital camera (Leica Microsystems, Milton Keynes, UK) and Mosaic imaging software (Tucsen V2.4). Images were captured as uncompressed JPEG files for optimal image quality.

Images were scored independently by two people, with labeling intensity of the tumor or normal epithelial cells in the images graded on a scale from 0 (no labeling) to 3 (strong labeling). 

### 2.6. Cell Knockdown

The shVPS4A (#VB181214–1076rkx) and shScrambled (SCR; #VB181214–1079yke) shRNA plasmids used in this study were designed using the Vector Builder online tool (www.vectorbuilder.com) and purchased from Vector Builder (Chicago, IL, USA). Plasmids were provided as E.coli stocks and plasmid extraction was performed using GenElute™ Plasmid Midiprep Kit (Merck). Next, optimum plasmid concentration was determined and verified by immunoblotting. Cells were transfected with 500 or 250 ng of shRNA, respectively, for 48 h using TransIT-X2^®^ Dynamic Delivery System (Mirus, Cambridge, UK). Later, stable transfection was obtained using puromycin selection, where cells were treated with 0.6 ug/mL of puromycin for 5 days [19]. Next, clonal selection was performed using a standardized dilution protocol [20] where a cell suspension of ≤ 10 cells/mL was prepared, and 100 µL of that was seeded into each well of a 96-well culture plate, which was then incubated at 37 °C in 5% CO_2_ for 2 weeks or until visible colonies were formed. Where wells contained single colonies, these were then collected, expanded, and tested for protein expression using immunoblotting, and the clones which exhibited the largest knockdown of VPS4A expression were selected for use in the experiments.

### 2.7. EGFR Assay

To confirm the function of the ESCRT complex, an EGFR assay where lysosomal degradation of epidermal growth factor receptor (EGFR) is monitored following induction by the agonist human epidermal growth factor (EGF) was carried out as described previously [21]. SW480/shVPS4A and SW480/shSCR cells were seeded in six-well plates at a density of 4 × 10^5^ cells/ well, and left to adhere overnight. The next day, cells were serum-starved for 24 h and then incubated with 50 ng/mL of human EGF (Thermofisher Scientific) for 0, 15, 30, 60, 90, or 120 min. The media was then removed and cells were collected by scraping directly into loading buffer and then separated by SDS-PAGE. Blots were probed with rabbit monoclonal anti-EGFR antibody (#ab52894, abcam) at 1:1000 dilution and with mouse monoclonal anti-GAPDH antibody (Proteintech) to ensure equal loading. Densitometric analysis was then carried out to monitor the levels of EGFR degradation over time [21].

### 2.8. Development of Oxaliplatin-Resistant Human CRC Sublines

CRC sublines with resistance to Oxa were derived and established from SW480 and KM12 wild-type (WT) parent cell lines by continuous exposure to increasing concentrations of Oxa over a period of ten months. Half maximal inhibitory concentration (IC_50_) values were used as initial starting doses for SW480 and KM12 parent cell lines, 0.7 and 2.5 μM, respectively. Oxa concentration was gradually increased up to 12.0 μM in the Oxa-resistant SW480 cell line (SW480/OxaR) and up to 42.0 μM in the Oxa-resistant KM12 cell line (KM12/OxaR). For each parent cell line, two controls grown in drug-free media were harvested in parallel for further analyses at low (SW480/WT P10 and KM12/WT P6) and high (SW480/WT P44 and KM12/WT P37) passages.

### 2.9. Chemosensitivity Assay

Chemosensitivity was evaluated using the MTT assay as described previously [22]. Briefly, 180 μL of 1 × 10^4^ cells/mL suspension was added to each test well of a 96-well plate and incubated overnight at 37 °C. Drugs or control solutions were added to each well, and the plates cultured under standard conditions for 4 days after which cells were incubated with MTT solution (5 mg/mL) in PBS for 4 h. Formazan crystals were then solubilized in 150 µL of DMSO (Merck) and the plates scanned at 540 nm using an Ao Microplate Reader (Azure Biosystems, Cambridge Bioscience Ltd, Cambridge, UK). Chemosensitivity in terms of the IC_50_ values was then determined from the data.

### 2.10. RNA Sequencing

RNA sequencing service and bioinformatic analysis was provided by NOVOGENE (Cambridge, UK). In short, mRNA was isolated from total RNA using magnetic beads attached to poly-T oligonucleotides. Following fragmentation, first-strand cDNA was synthesized with random hexamer primers, and subsequently, the second strand of cDNA was produced. The quality of the library was assessed using Qubit for quantification, real-time PCR for further validation, and a bioanalyzer to evaluate size distribution. The quantified libraries were then pooled and sequenced on a NovaSeq 6000 S4 platform (Illumina). For differential expression analysis between groups, the data were analyzed using the DESeq2 R package (version 1.48.1) [23].

### 2.11. Statistical Analysis

All statistical tests for MTT assays and protein expression fold change were generated using GraphPad Prism 9.0 (GraphPad Software, Inc., San Diego, CA, USA). Results are expressed as the means ± SEM of 3 independently repeated experiments. An Unpaired *t*-test was used for statistical analyses, with significance scored at * *p* ≤ 0.05, ** *p* ≤ 0.01 and *** *p* ≤ 0.001 levels, and any *p* value > 0.05 was considered not significant (ns). 

For monitoring of synergy in the combination studies for Oxa and aloperinol, the survival data from the study was analyzed using Synergy Finder software version 3.0, (synergyfinder.fimm.fi) with the expected drug combination responses calculated based on the ZIP reference model. Deviations between observed and expected responses with positive and negative values denote synergy and antagonism, respectively [24].

## 3. Results

### 3.1. VPS4A Is Highly Expressed Across a Panel of CRC Cell Lines

ESCRT proteins representing the different ESCRT subtypes were evaluated in a panel of 10 CRC cell lines using immunoblotting to characterize the baseline expression levels and select cell lines for manipulation of expression in further studies. Detection in the cancer cell lines was normalized to expression in a non-cancer intestinal epithelial cell line, HIEC-6. Immunoblots (Figure 1a) and a heat map created from the densitometry analysis (Figure 1b) exhibited a range of expression levels, with two ESCRT proteins relatively consistently overexpressed across the whole of the cell line panel: CHMP6 and VPS4A. Given the more comprehensive literature links of VPS4A and cancer and findings in previous in-house proteomic studies, as discussed below, we then focused our initial studies presented in this paper on VPS4A.

The antibody used to probe for VPS4A also detected VPS4B, and densitometric analysis showed that whilst there was significant change in the panel for VPS4A, no significant changes were seen for VPS4B (Figure 1c,d).

### 3.2. VPS4A Overexpressed in CRC Samples

It was observed that VPS4A was highly expressed in the tumor epithelium for all nine patients compared to the surrounding tissue, with immunolabeling seen in the cytoplasmic compartment. For patients where normal intestinal epithelium was also present in the samples, it was seen that there was a lower expression of VPS4A in this tissue compared to the areas of the tumor (Figure 2).

When labeling intensity was quantified, it was seen that there was a highly significant difference (*p* < 0.001) between the mean grade for tumor epithelium (mean grade 2.9 ± 0.2) compared to normal intestinal epithelium (mean grade 0.5 ± 0.2).

### 3.3. Knockdown of VPS4A Expression Leads to Loss of ESCRT Function

Following confirmation that VPS4A was relatively highly expressed in the wild-type CRC cell line panel, the SW480 and KM12 cell lines were selected for further studies due to the significant overexpression and suitability for transfection [25]. Immunoblotting confirmed establishment of a subline clone with constitutively knocked down VPS4A expression for both SW480 (clone 4) and KM12 (clone 5), with densitometry analysis demonstrating significantly reduced levels. In contrast, the analysis demonstrated that levels of the closely related VPS4B remained the same (Figure 3a,b).

Corroboration that this knockdown led to loss of ESCRT machinery function using the EGFR assay demonstrated the rate of degradation of EGFR was slowed in the SW480/shVPS4A knockdown clone 4 and KM12/shVPS4A clone 5 sublines compared to the SW480/shSCR and KM12/shSCR control sublines, respectively (Figure 3c,d).

### 3.4. Reduction of VPS4A Expression Sensitizes Cells to Exposure to Oxaliplatin and Other Commonly Used Treatments for CRC

To determine if modulating expression of VPS4A by knocking down expression in CRC cell line could improve sensitivity to agents usually used in the treatment of CRC, Oxa, 5-FU, and Iri, the SW480/shVPS4A and KM12/shVPS4A knockdown sublines were treated with each agent and response compared to treatment of the SW480/shSCR and KM12/shSCR control sublines, respectively, using an MTT cytotoxicity assay. Reducing VPS4A expression significantly increased sensitivity to Oxa. Whilst response to 5-FU did not change significantly in both cell lines, response to Iri was significantly altered in KM12 (Table 2 and Appendix A).

### 3.5. VPS4A Expression Is Altered in Oxa-Resistant Cell Lines

After it was seen that modulating VPS4A expression influenced Oxa sensitivity, correlation of VPS4A expression with sensitivity to Oxa was explored. To this end, we developed two Oxa-resistant cell lines derived from SW480/WT and KM12/WT CRC cell lines, with 5.6-fold and 18.6-fold resistance to Oxa, respectively (Figure 4a,b). We consequently evaluated VPS4A expression in the cell lines using immunoblotting and found that by densitometry, it was significantly higher in the Oxa-resistant cell lines (SW480/OxaR and KM12/OxaR) compared to the Oxa-sensitive WT lines (Figure 4c,d) (*p* < 0.001 and *p* < 0.05 for SW480 and KM12, respectively), thus confirming the potential modulatory role in Oxa resistance.

When VPS4A expression was subsequently knocked down in the SW480/OxaR subline (SW480/OxaR shVPS4A), as confirmed by immunoblotting (Figure 5a,b), decreased functionality of the protein was confirmed by reduced EGFR degradation in the EGFR assay compared to the SW480/OxaR shSCR cell line (Figure 5c,d). When both lines were treated with Oxa and analyzed by the MTT assay, it was seen that sensitivity to Oxa returned in the SW480/OxaR shVPS4A line, adding further support to VPS4A’s role in resistance modulation (Figure 5e).

### 3.6. Aloperine, an Inhibitor of VPS4A, Modulates Resistance to Oxaliplatin in Both Wild-Type and Oxa-Resistant Cell Lines

To demonstrate the potential of using therapeutic control of VPS4A levels to modulate Oxa sensitivity in CRC cell lines, we utilized a molecule previously known to inhibit VPS4A expression, aloperine [26]. We first confirmed, using immunoblotting, that it inhibited VPS4A. Some inhibition of VPS4B was also detected but to a lesser extent in SW480 cells (Figure 6a).

The next step was to determine if reduction of VPS4A expression with Alo could modulate sensitivity to Oxa. We exposed SW480/WT and SW480/OxaR cell lines with combinations of Oxa and Alo for 96 h and assessed cytotoxicity using the MTT assay as described above. The data was then analyzed using SyndergyFinder, and it was seen that a synergistic relation was seen in the OxaR cell line, suggesting that the Alo had potentiated the effect of Oxa to overcome resistance, whilst in the WT cell line, an additive effect was observed (Figure 6b, and Table 3).

### 3.7. Decreased Expression of the Drug Efflux Transporter MRP2 (ABCC2) as a Result of the Knockdown of VPS4A 

The next steps were to gain information on a potential mechanism of action by which VPS4A could modulate oxaliplatin resistance. A genomic approach was taken with RNA sequencing, used to highlight differences in gene expression between SW480/shVPS4A and SW480/shSCR samples (see Appendix A for gene sequencing data). Genes were ranked in order of differential expression between scrambled and VPS4A knockdown cell lines, and the data was analyzed, looking for genes whose protein product was known to be involved in Oxa resistance [3,27,28,29]. The *ABCC2* gene, which encodes for the Multiple Resistance Protein 2 (MRP2), was seen to be the most relevant differentially expressed gene from the data, with an approximately two-fold reduction in gene expression in the VPS4A knockdown subline (Figure 7a). Subsequently we confirmed that the reduction in *ABCC2* expression translated to a reduction in MRP2 expression by immunoblotting in both the SW480/shVPS4A and SW480/ OxaR/ shVPS4A sublines compared to their respective scrambled sublines (Figure 7b,c). Thus, this would suggest that VPS4A may modulate Oxa sensitivity by controlling MRP2 expression to decrease Oxa efflux from the cell.

## 4. Discussion

Drug resistance is still a significant barrier to improving disease-free survival rates for CRC patients, and there is considerable work still being carried out to identify novel mechanisms of drug resistance to develop predictive biomarkers for optimizing treatment regimens and to develop modulators which could be administered to overcome treatment resistance. Approaches to modulate resistance include targeted protein degradation, combining antibody–cytotoxic drug conjugates with immune checkpoint inhibitors, and targeting the tumor microenvironment [30].

In this study, compelling evidence to support the ESCRT accessory protein VPS4A as a modulator of drug resistance in CRC is presented. Whilst there have been studies which suggest that ESCRT family proteins, in general, and VPS4A, in particular, have a role in cancer processes and resistance [10,16,31], this is the first study, to our knowledge, where a direct link between modulation of VPS4A expression and resistance to drugs in CRC cell lines has been demonstrated. The few only other published studies which talk of direct involvement of ESCRT family members in cancer resistance mechanisms involve CHMP2A, which affects the sensitivity of cancer cells to natural killer-cell-mediated cytotoxicity in glioblastoma and head and neck squamous cell carcinoma which might participate in the development of drug resistance [17]. 

This study initially looked to characterize ESCRT expression in a panel of ten CRC cell lines with a range of phenotypes to identify ESCRT family members which were either consistently overexpressed or underexpressed across the panel. The aim was to select a candidate which could be a factor in resistance across as many CRC subtypes as possible, not solely focusing on one phenotype, and to give the best chance to be effective in either a diagnostic or therapeutic role going forward. Whilst a heterogeneity of expression for the different ESCRTs across the panel was to be expected from observations of ESCRT expression in CRC cell lines on the Protein Atlas Database (www.proteinatlas.org), two candidates emerged from these initial studies, CHMP6 and VPS4A. The latter was selected for further analysis given that there is a more comprehensive literature covering links for VPS4A and cancer [16,31,32], whilst in the case of CHMP6, only a few papers have demonstrated links to cancer [33,34]. In addition, previous in-house proteomic studies have demonstrated across the board upregulation of VPS4 protein in CRC drug-resistant cell lines, with the 5-FU-focused data published previously [22]. 

Two lines, KM12 and SW480, were selected for further investigation, as they were the lines with the highest overexpression of VPS4A. Both these cell lines share some similarities in their genetic profile, with both being mutant for p53 [35,36]. Whilst there are no reports in the literature of a direct link between p53 and VPS4A expression, interestingly, involvement of p53 in the regulation of the ESCRTIII protein CHMP4C has been reported, indicating cross talk between both the p53 and ESCRT pathways, which may have an effect on VPS4A overexpression [37].

VPS4A elevated expression was not only demonstrated in a cell line panel, but also translated to clinical material, where raised VPS4A expression was observed in malignant transformed epithelium compared to adjacent normal intestinal epithelium in a small sample of CRC clinical material, thus showing the clinical relevance of this protein. 

The key findings of the study came next, where it was demonstrated that by knocking down expression of VPS4A in a CRC cell line, its function could be modulated and ultimately sensitize the cells to CRC drug treatment, significantly in the case of Oxa, with some sensitization also evident to a lesser extent for Iri and 5-FU. This fits with similar findings where sensitivity to platinum drugs has been improved following modulation of other intracellular transport mechanisms. For example, enhancing sensitivity of CRC cells to Oxa through inhibition of transporter molecule MRP2 by dihydromyricetin [38] or siRNA knockdown of the *ABCC2* gene encoding MRP2 [39] enhanced CRC cells’ sensitivity to Oxa.

Further validation of our VPS4A hypothesis came in studies where it was seen that VPS4A expression was increased in Oxa-resistant sublines for a couple of CRC lines, and in further studies, it was demonstrated that knocking down VPS4A expression in the SW480/OxaR cell line could reverse Oxa resistance.

We envisage that the evidence collected in this study will be utilized to develop a specific antagonist of VPS4A which can be administered from the start of chemotherapy with Oxa to prevent tumor cells becoming resistant to the drug and improve the chances of success with it. This approach has been previously taken with other potential resistance modulators from different mechanisms; for example, in studies combining the DNA methyltransferase inhibitor decitabine [40], the flavone glycoside scutellarin [41], or Kruppel-like factor 5 inhibitor, ML264 [42], with Oxa in CRC cell lines or patient-derived organoids.

With this in mind, a small molecule, Alo [26], which was identified as a novel autophagy inhibitor that triggers tumor cell death by targeting VPS4A in both in vitro and in vivo non-small cell lung cancer models, was utilized. We demonstrated synergism when administered along with Oxa in the SW480/OxaR cell line, suggesting the validity of this approach. However, given the high doses of Alo which were required to demonstrate VPS4 knockdown, and the questionable VPS4A specificity, with a clear knockdown of VPS4B seen as well, a more specific and potent molecule would be required for an effective strategy. Therefore, the next step will be to carry out a drug discovery program based on this molecule to optimize its specificity and drug-like properties. A previous study by Huang et al. [10] identified druggable structures within the VPS4A molecule, and targeting these will also be incorporated into the drug discovery program. We will also look to extend our investigation into assessing the other promising ESCRT family member in terms of expression, CHMP6.

In determining the mechanism by which VPS4A may have an effect on modulation of Oxa sensitivity, it was interesting that the expression of the *ABCC2* gene was significantly repressed when RNA sequencing studies were carried out, as reported above. Additionally, some of the genes involved in regulation of MRP2, such as NR1H2 and NR1H3 [43], were also seen to have a decreased expression. Potentially, this could explain the re-sensitization of the colon cancer cells to Oxa after VPS4A knockdown. However, the exact mechanism by which VPS4A has influence on control of MRP2 expression still needs further investigation.

## 5. Conclusions

In this study, it was demonstrated that the ESCRT family member VPS4A is a potential target for the modulation of Oxa resistance in CRC, both through knockdown of VPS4A expression in CRC cell lines and by chemical inhibition. The evidence presented has led us to hypothesize that the mechanism for the modulation of Oxa sensitivity may come through control of MRP2. The next steps will be to look to develop a pharmacologically active and effective inhibitor of VPS4A which can be utilized to improve the outcome of Oxa-based chemotherapy in CRC patients, and to further explore the mechanism of modulation.

## Figures and Tables

**Figure 1 cells-14-00929-f001:**
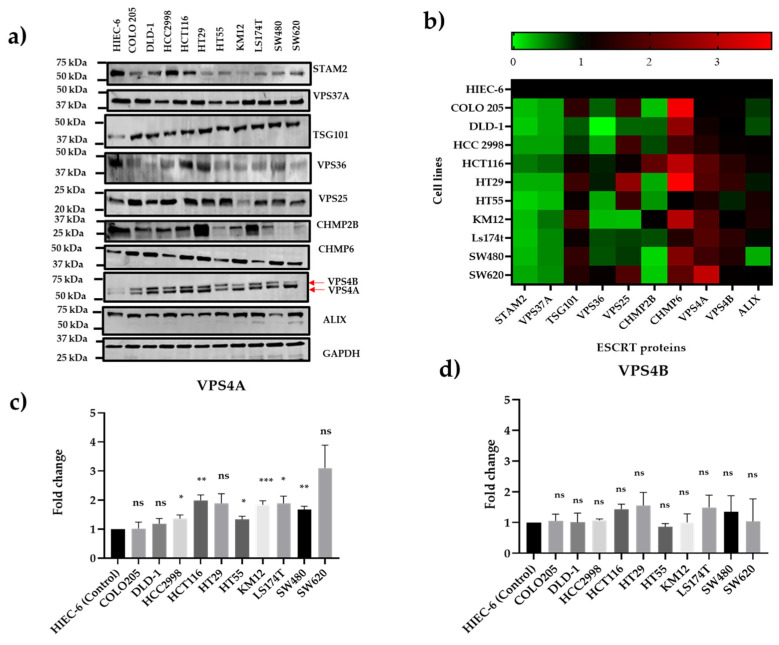
Characterization of ESCRT protein expression in a panel of human CRC cell lines using immunoblotting. (**a**) Immunoblot and (**b**) heat map showing the relative expression of the ESCRT proteins compared to expression in a non-cancer intestinal epithelial cell line, HIEC-6, for three separate immunoblot runs. Heat map scores: (green) = levels lower (0), (black) = equivalent (1), and (red) = greater expression (2/3) than HEIC-6 cells. Of interest is that VPS4A is one of only two ESCRT proteins which are consistently overexpressed in the panel, the other being CHMP6. (**c**,**d**) Densitometric analysis for VPS4A and VPS4B protein expression showing that whilst significant high expression is seen for a range of the cell lines for VPS4A, none of the cell panels show a significantly high expression for VPS4B. Statistical differences compared to HIEC-6 are highlighted by * *p* < 0.05, ** *p* < 0.01, and *** *p* < 0.001 using Student’s *t*-test.

**Figure 2 cells-14-00929-f002:**
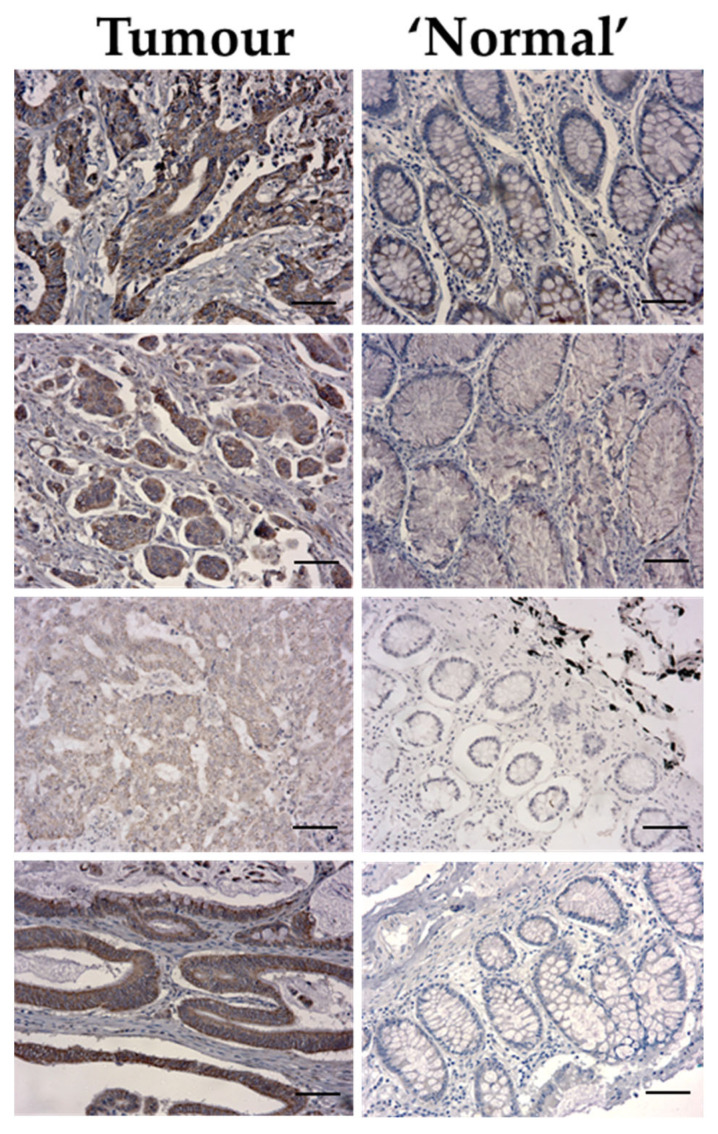
Representative images of VPS4A immunolabeling for 4 different patients showing higher expression in the cytoplasmic fraction of the tumor epithelium compared to the normal epithelium. Bar length = 60 µm.

**Figure 3 cells-14-00929-f003:**
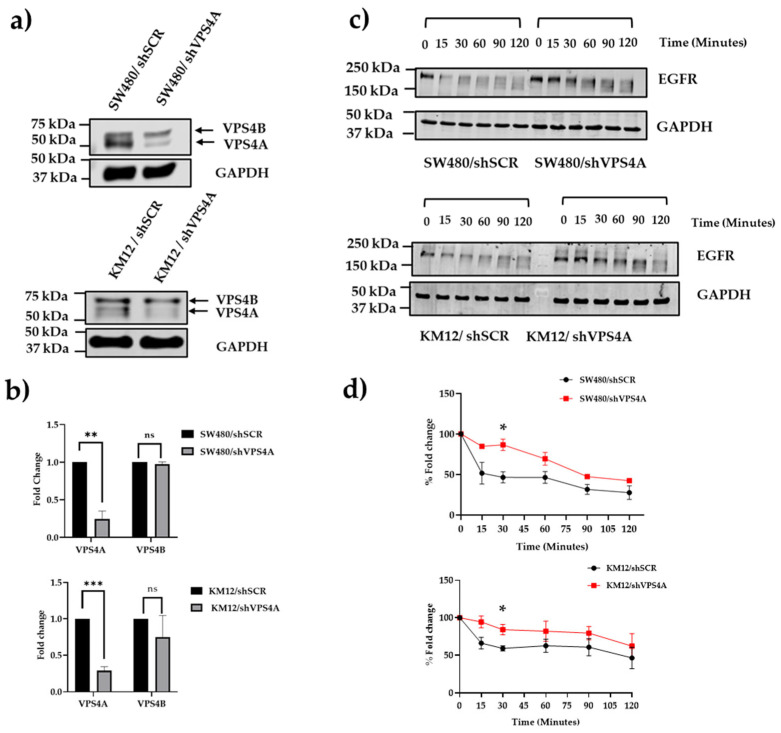
(**a**,**b**) Subline of SW480 (SW480/shVPS4A) and KM12 (KM12/shVPS4A) with constitutively lower expression of VPS4A were established as confirmed using immunoblotting (n = 3). Expression of VPS4B, in contrast, was not significantly altered. Statistical differences compared to the respective scrambled subline are highlighted by ** *p* < 0.01 and *** *p* < 0.001 using Student’s *t*-test. (**c**,**d**) Loss of ESCRT machinery function as a consequence of this knockdown was confirmed for both SW480 and KM12 sublines using the EGFR assay, with analysis using immunoblotting (n = 3), where there was a significant (* *p* < 0.05 using Student’s *t*-test) slowing in the rate of EGFR degradation compared to the scrambled control sublines (SW480/shSCR and KM12/shSCR) where the ESCRT machinery was intact. EGFR protein expression in (**d**) is normalized to t = 0.

**Figure 4 cells-14-00929-f004:**
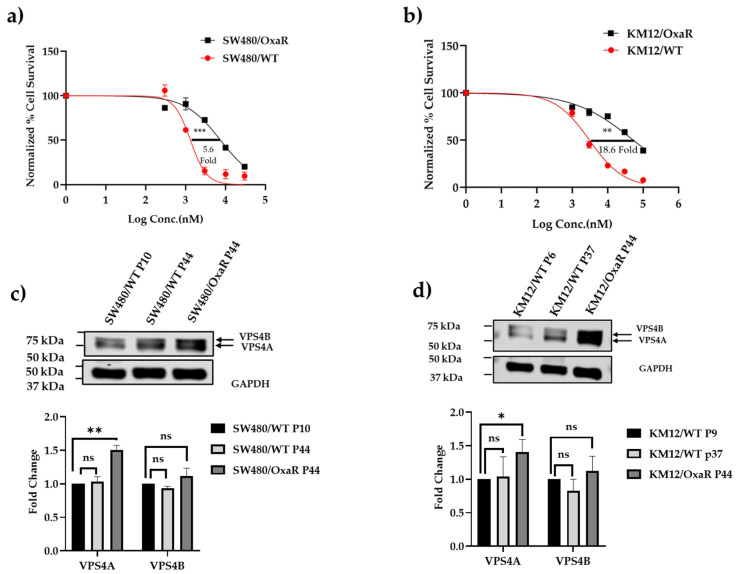
(**a**,**b**) Generation of Oxa-resistant cell lines demonstrating significant resistance to the drug compared to the wild-type cell line in an MTT chemosensitivity assay. Cell survival profiles of parent cell lines and their respective resistant sublines to Oxa for a) SW480 and b) KM12 CRC cell lines under exposure to 0.3–100 µM doses of Oxa for 96 h. MTT assays were performed in three independent experiments. A 5.6-fold and 18.6-fold difference in IC_50_ were found in the SW480/OxaR *** *p* < 0.001 and KM12/OxaR ** *p* < 0.01 sublines, respectively. (**c**,**d**) demonstrate a significantly raised expression of VPS4A in the resistant cell lines respective to their parent wild-type cell lines using immunoblotting (*n* = 3). The expression of VPS4B, in contrast, was not significantly altered. Statistical differences compared to the wild type are highlighted by * *p* < 0.05 and ** *p* < 0.01 using an Unpaired Student’s *t*-test.

**Figure 5 cells-14-00929-f005:**
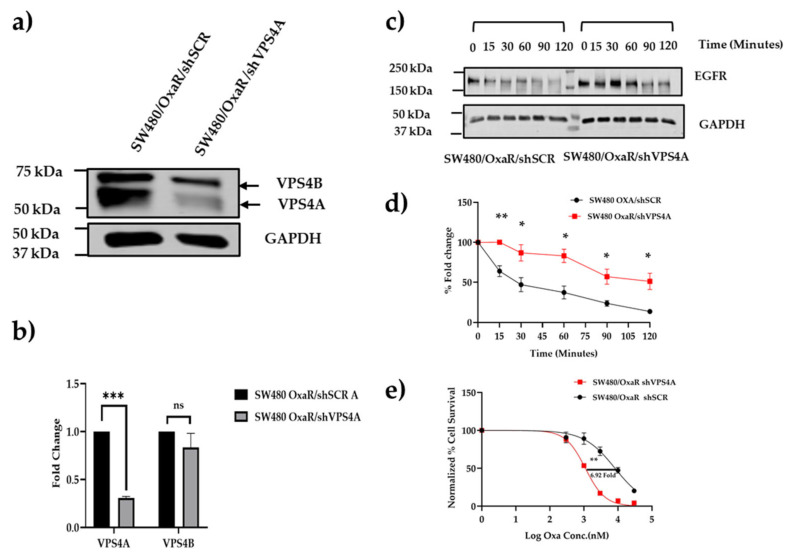
(**a**,**b**) A subline of SW480/OxaR with constitutively lower expression of VPS4A was established as confirmed using immunoblotting (*n* = 3); statistical differences compared to the scrambled subline are highlighted by *** *p* < 0.001 using Student’s *t*-test. The expression of VPS4B, in contrast, was not significantly altered. (**c**,**d**) Loss of ESCRT machinery function as a consequence of this knockdown was confirmed using the EGFR assay, with analysis using immunoblotting (*n* = 3), where there was a significant slowing (* *p* < 0.05 and ** *p* < 0.01, using Student’s *t*-test) in the rate of EGFR degradation compared to the scrambled subline where the ESCRT machinery was intact. EGFR protein expression is normalized to the control at t = 0. (**e**) Knocking down VPS4A expression in the SW480/OxaR subline reverses the Oxa resistance phenotype, as measured in an MTT cytotoxicity assay (*n* = 3). Statistical differences between the knockdown and scrambled cell lines are highlighted by ** *p* < 0.01 using Student’s *t*-test.

**Figure 6 cells-14-00929-f006:**
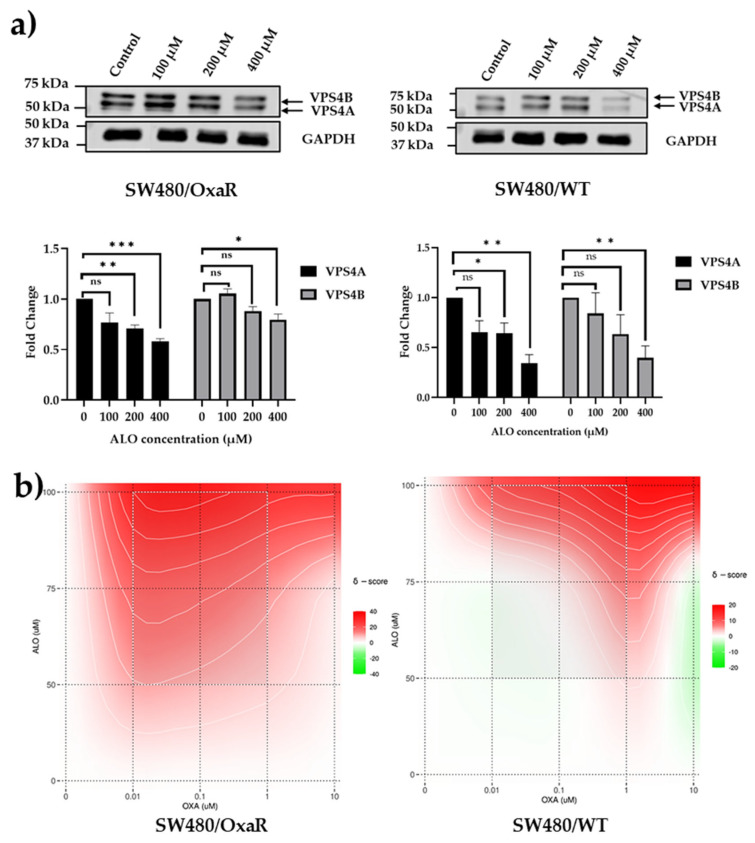
(**a**) Aloperine inhibits expression of VPS4 proteins. SW480 cells were treated with Alo for 96 h and then harvested and analyzed using immunoblotting. (**b**) Representative contour plots following administration of combinations of Oxa and Alo to SW480/WT and SW480/OxaR cell lines for 96 h in the MTT assay, showing more synergistic interactions between concentrations of each compound for the SW480/OxaR cell line compared to the SW480/WT cell line, as denoted by the higher area of the plot shaded in red. Statistical differences compared to untreated controls are highlighted by * *p* < 0.05, ** *p* < 0.01, and *** *p* < 0.001 using Student’s *t*-test.

**Figure 7 cells-14-00929-f007:**
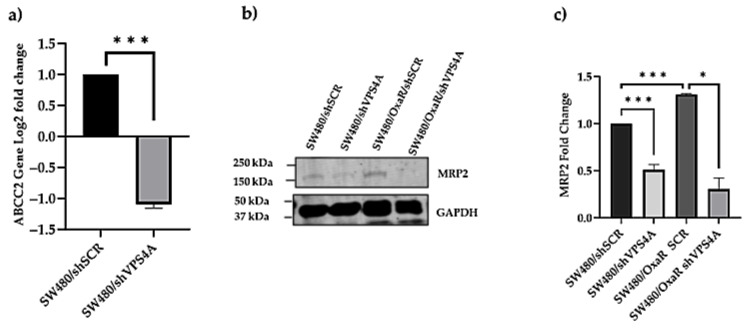
(**a**) Changes in *ABCC2* gene expression when VPS4A is modulated. A statistically significant change in *ABCC2* gene expression was seen for the SW480/shVPS4A subline compared to the scrambled control SW480/shSCR in RNA sequencing data (*n* = 3). (**b**,**c**) MRP2 protein expression fold change in SW480/shVPS4A and SW480 OxaR shVPS4A compared to the respective scrambled controls using immunoblotting (*n* = 3). Statistical differences compared to the scrambled subline are highlighted by * *p* < 0.05 and *** *p* < 0.001 using Student’s *t*-test.

**Table 1 cells-14-00929-t001:** Details for the colorectal cancer patients included in the study.

Patient ID Code	Sex	Stage T	Stage N	Stage M
9	M	4	1	1
22	M	3	0	1
24	F	4	2	1
31	M	3	2	1
33	M	3	1	1
37	F	3	2	1
56	M	3	1	1
57	F	4	0	1
58	F	3	0	1

Abbreviations: ID—identification; M—male, and F—female; Stage T—size and extent of the primary tumor; Stage N—extent of spread to regional lymph nodes; Stage M—tumor has metastasized.

**Table 2 cells-14-00929-t002:** Knocking down VPS4A expression significantly increased sensitivity to Oxa in the SW480 and KM12 cell lines, with variable effects in combination with other agents commonly used in the treatment of CRC in an MTT cytotoxicity assay (*n* = 3). Statistical differences between the knockdown and scrambled cell lines are highlighted by * *p* < 0.05 and ** *p* < 0.01 using Student’s *t*-test.

Drug	IC_50_ (µM) for SW480/shVPS4A	IC_50_ (µM) for SW480shSCR	Fold-Difference	*p* Value
Oxaliplatin	0.47 ± 0.05	1.6 ± 0.17	3.4	**
5-Fluorouracil	1.73 ± 0.07	2.37 ± 0.3	1.37	ns
Irinotecan	4.23 ± 0.93	6.56 ± 1.35	1.55	ns
**Drug**	**IC_50_ (µM) for KM12/shVPS4A**	**IC_50_ (µM) for KM12/shSCR**	**Fold-difference**	***p* value**
Oxaliplatin	1.53 ± 0.12	2.17 ± 0.17	1.44	*
5-Fluorouracil	2.17 ± 0.26	2.13 ± 0.43	1.05	ns
Irinotecan	2.50 ± 0.12	9.17 ± 0.44	3.67	**

**Table 3 cells-14-00929-t003:** Mean synergy scores for 3 independent runs for both cell lines, showing that the interaction between Alo and Oxa is more likely to be synergistic in the OxaR cell line, whilst the interactions are additive in the WT cell line.

	SW480/OxaR	SW480/WT
**Synergy Score**	15.61 ± 1.16	3.00 ± 0.99
**Interpretation**	Likely to be synergistic	Likely to be additive

## Data Availability

The original contributions presented in this study are included in the article/Appendix A. Further inquiries can be directed to the corresponding author (S.D.S.).

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
