# Peer review of "A Novel Modulator of Resistance for Oxaliplatin-Based Therapy for Colorectal Cancer: The ESCRT Family Member VPS4A"

_cells, 2025, doi:10.3390/cells14120929_

Round 1
Reviewer 1 Report
Comments and Suggestions for Authors
The authors have identified VPS4A as a molecule involved in resistance to Oxalipatin. While the observation is important, the mechanism causing the selective resistance to Oxaliplatin needs further investigation. The publication would not hold enough merit without delineating the underlying mechanism. Further functional observation was only tested in a single cell line. At least 2 of the cell lines mentioned in the publication need to have the complete set of data for the knockdown and drug combination studies.
Major comment:
Hypothesize and provide evidence for the mechanism of resistance mediated by the VPS4A protein to Oxaliplatin.
Complete the main data sets with KM12 and KM12 OXAR cells
Minor comments
Figure 1:It is unclear why the mRNA levels of VSP4A do not show any significant difference. Is this a post-translational regulation?
Please explain each subfigure in the text.
Figure 2: Quantify IHC figures.
Figure 3. Please explain each subfigure in the text and provide an adequate explanation.
Table 3: Show MTS assay curves
Table 4: Show MTS assay curves.
Comments on the Quality of English LanguageThe figures need to be properly explained in the text.
Author Response
We thank the reviewer for their constructive and helpful feedback and have addressed the revisions they have suggested as covered below:
Major comment:
Hypothesize and provide evidence for the mechanism of resistance mediated by the VPS4A protein to Oxaliplatin.
We have now included a hypothesis for the mechanism of resistance based on RNA sequencing data where we have compared RNA expression in the SW480/SCR and /VPS4A knockdown cell lines. Here we looked for genes whose expression was significantly reduced in the knockdown line where there was a known linkage to involvement in Oxaliplatin resistance (please see section 3.7), and it was seen that expression of the ABCC2 gene, which encodes for the drug efflux transporter MRP2 was significantly reduced. We hypothesize from this that the mechanism of action of VPS4A modulation is through depleting levels of the drug efflux transporter MRP2 in the cell, preventing oxaliplatin egress and increasing cell exposure to the drug.
Complete the main data sets with KM12 and KM12 OXAR cells
We have included the KM12 data sets (please see sections 3.3 and 3.4)
Minor comments
Figure 1:It is unclear why the mRNA levels of VSP4A do not show any significant difference. Is this a post-translational regulation?
We are unsure what the reviewer is alluding to here as we only cover protein and not mRNA expression levels in this Figure.
Please explain each subfigure in the text.
We have adjusted the main body of text in the results section accordingly.
Figure 2: Quantify IHC figures.
We have now carried out quantification and modified sections 2.5 and 3.2 accordingly.
Figure 3. Please explain each subfigure in the text and provide an adequate explanation.
We have adjusted the main body of text in the results section accordingly.
Table 3: Show MTS assay curves
We have included the MTT (not MTS) assay curves as a supplementary figure and amended Table 3 to include the KM12 data
Table 4: Show MTS assay curves.
We have replaced Table 4 with the MTT (not MTS) assay curves in Figure 5 (e)
Comments on the Quality of English Language
The figures need to be properly explained in the text.
As mentioned above, we have adjusted the main body of text in the results section accordingly.
Reviewer 2 Report
Comments and Suggestions for Authors
CRC remains one of the most important tumoral pathologies. Chemotherapy remains the backbone of treatment, so studies related to resistance or increasing efficacy continue to be of interest.
Resistance or efficacy from the macroscopic aspect should be translated into the molecular aspects in order to increase the chance of adapting the therapy better to the continuous changes of the tumour.
The article has the merit to do that, so I think that it is a good scientifical acquisition for the journal.
Author Response
We thank the reviewer for their constructive and helpful feedback
Reviewer 3 Report
Comments and Suggestions for Authors
cells-3673677
Type: Article
Title: A novel modulator of resistance for oxaliplatin-based therapy for colorectal cancer: the ESCRT Family Member VPS4A
Authors: Noha M Abdelrazik , Anjana Patel , Andrew Conn , Christopher W Sutton , Sriharsha Kantamneni , Steven D Shnyder *
This study presents a novel insight into the role of the ESCRT family protein VPS4A in mediating resistance to oxaliplatin (Oxa) in colorectal cancer (CRC). While drug resistance remains a long-standing barrier to effective chemotherapy in CRC, the involvement of intracellular trafficking mechanisms, particularly the ESCRT machinery, is underexplored. This research fills a critical gap by proposing VPS4A as a modulator of Oxa resistance, and introduces aloperine, a small molecule VPS4A inhibitor, as a potential sensitizing agent. However, before the paper is accepted, the following points should be additionally corrected:
[Major concerns]
- Clarify Mechanism: Investigate how VPS4A mechanistically affects oxaliplatin transport, DNA damage response, or apoptosis pathways. Consider including RNA-seq or phosphoproteomics.
- Introduce In Vivo Experiments: Include xenograft or PDX models to evaluate whether VPS4A knockdown or inhibition enhances oxaliplatin efficacy in vivo.
- Validate Aloperine Specificity: Perform target engagement studies (e.g., CETSA, SPR, or pull-down assays) to confirm direct interaction between aloperine and VPS4A.
- SW480 cells: The authors said that among the 10 colon cancer cells used in this study, SW480 colon cancer cells showed a special response to this study, so they performed additional experiments. However, there is no explanation at all as to why SW480 colon cancer cells showed such responses. In general, in studies using cancer cells, it may be due to the characteristics of specific oncogenes, tumor suppressor genes, or other genes, so I hope the authors will discuss this part in the Discussion section.
- English: I feel embarrassed to point out English-related issues in a paper submitted by British confectioners. However, I cannot help but make a few points about English writing. 1) In general, protein or compound names are not necessarily capitalized in the middle of a sentence. In this paper, there are many cases where the first letter of the word for protein or compound is capitalized when it comes to abbreviations, so please correct them all. 2) There are cases where specific cell names are written differently, so please correct them. Examples: HT55 at Line 84 vs. HT-55 at Line 87; KM12 vs. KM-12; etc.
- Abbreviations: The use of abbreviations when writing a paper has many advantages besides simplicity of expression. To use an abbreviation, first write the abbreviation in parentheses after the full name, and then use the abbreviation from Introduction to the final Conclusion. Abbreviations should only be used if they are repeatedly used and if they are not used again, only the full name should be used. Whether it's an abstract or a text, systematically proofread each abbreviation to make sure there are no abbreviations that have been redefined repeatedly.
- In cases where abbreviations are used within figures or tables, please list these abbreviations along with their corresponding full names in the figure legends or at the bottom of corresponding tables. If there are two or more abbreviations, arrange them in alphabetical order.
- Materials and Methods section - When naming a particular chemical company, you must provide location information such as company name, city, and/or state (abbreviation in the USA and Canada) and country. Once you have named a company with the information, you should only mention a company’s name thereafter.
- Notation of units: In scientific papers, almost all units, except for temperature and percentage, are written with a space between the number and the unit. However, in this paper, numbers and units are written together without spaces in both the figures and the text, which is very distracting.
- Figure 2: Typically, each photo is accompanied by a bar scale; however, in this instance, only the final photo features one.
[Minor concerns]
- Figure 1a: ‘KDa’ should be written as ‘kDa’.
- Table 3: IC50 should be written as IC50.
- Line 429: In the case of human genes, they should be written in italics.
- References: Adjust the citation style of the references to conform to the cells guidelines, and ensure that any missing volume and page numbers are accurately included. Examples: 1, 2, 3, 5, 6, 7, 22, 32, 33, etc.
Overall, the manuscript can be considered to publication after major revision as indicated above.
Comments on the Quality of English Languagecells-3673677
Type: Article
Title: A novel modulator of resistance for oxaliplatin-based therapy for colorectal cancer: the ESCRT Family Member VPS4A
Authors: Noha M Abdelrazik , Anjana Patel , Andrew Conn , Christopher W Sutton , Sriharsha Kantamneni , Steven D Shnyder *
This study presents a novel insight into the role of the ESCRT family protein VPS4A in mediating resistance to oxaliplatin (Oxa) in colorectal cancer (CRC). While drug resistance remains a long-standing barrier to effective chemotherapy in CRC, the involvement of intracellular trafficking mechanisms, particularly the ESCRT machinery, is underexplored. This research fills a critical gap by proposing VPS4A as a modulator of Oxa resistance, and introduces aloperine, a small molecule VPS4A inhibitor, as a potential sensitizing agent. However, before the paper is accepted, the following points should be additionally corrected:
[Major concerns]
- Clarify Mechanism: Investigate how VPS4A mechanistically affects oxaliplatin transport, DNA damage response, or apoptosis pathways. Consider including RNA-seq or phosphoproteomics.
- Introduce In Vivo Experiments: Include xenograft or PDX models to evaluate whether VPS4A knockdown or inhibition enhances oxaliplatin efficacy in vivo.
- Validate Aloperine Specificity: Perform target engagement studies (e.g., CETSA, SPR, or pull-down assays) to confirm direct interaction between aloperine and VPS4A.
- SW480 cells: The authors said that among the 10 colon cancer cells used in this study, SW480 colon cancer cells showed a special response to this study, so they performed additional experiments. However, there is no explanation at all as to why SW480 colon cancer cells showed such responses. In general, in studies using cancer cells, it may be due to the characteristics of specific oncogenes, tumor suppressor genes, or other genes, so I hope the authors will discuss this part in the Discussion section.
- English: I feel embarrassed to point out English-related issues in a paper submitted by British confectioners. However, I cannot help but make a few points about English writing. 1) In general, protein or compound names are not necessarily capitalized in the middle of a sentence. In this paper, there are many cases where the first letter of the word for protein or compound is capitalized when it comes to abbreviations, so please correct them all. 2) There are cases where specific cell names are written differently, so please correct them. Examples: HT55 at Line 84 vs. HT-55 at Line 87; KM12 vs. KM-12; etc.
- Abbreviations: The use of abbreviations when writing a paper has many advantages besides simplicity of expression. To use an abbreviation, first write the abbreviation in parentheses after the full name, and then use the abbreviation from Introduction to the final Conclusion. Abbreviations should only be used if they are repeatedly used and if they are not used again, only the full name should be used. Whether it's an abstract or a text, systematically proofread each abbreviation to make sure there are no abbreviations that have been redefined repeatedly.
- In cases where abbreviations are used within figures or tables, please list these abbreviations along with their corresponding full names in the figure legends or at the bottom of corresponding tables. If there are two or more abbreviations, arrange them in alphabetical order.
- Materials and Methods section - When naming a particular chemical company, you must provide location information such as company name, city, and/or state (abbreviation in the USA and Canada) and country. Once you have named a company with the information, you should only mention a company’s name thereafter.
- Notation of units: In scientific papers, almost all units, except for temperature and percentage, are written with a space between the number and the unit. However, in this paper, numbers and units are written together without spaces in both the figures and the text, which is very distracting.
- Figure 2: Typically, each photo is accompanied by a bar scale; however, in this instance, only the final photo features one.
[Minor concerns]
- Figure 1a: ‘KDa’ should be written as ‘kDa’.
- Table 3: IC50 should be written as IC50.
- Line 429: In the case of human genes, they should be written in italics.
- References: Adjust the citation style of the references to conform to the cells guidelines, and ensure that any missing volume and page numbers are accurately included. Examples: 1, 2, 3, 5, 6, 7, 22, 32, 33, etc.
Overall, the manuscript can be considered to publication after major revision as indicated above.
Author Response
We thank the reviewer for their constructive and helpful feedback and have addressed the revisions they have suggested as covered below:
[Major concerns]
- Clarify Mechanism: Investigate how VPS4A mechanistically affects oxaliplatin transport, DNA damage response, or apoptosis pathways. Consider including RNA-seq or phosphoproteomics.
We have now included a hypothesis for the mechanism of resistance based on RNA sequencing data where we have compared RNA expression in the SW480/SCR and /VPS4A knockdown cell lines. Here we looked for genes whose expression was significantly reduced in the knockdown line where there was a known linkage to involvement in Oxaliplatin resistance (please see section 3.7), and it was seen that expression of the ABCC2 gene, which encodes for the drug efflux transporter MRP2 was significantly reduced. We hypothesize from this that the mechanism of action of VPS4A modulation is through depleting levels of the drug efflux transporter MRP2 in the cell, preventing oxaliplatin egress and increasing cell exposure to the drug.
- Introduce In Vivo Experiments: Include xenograft or PDX models to evaluate whether VPS4A knockdown or inhibition enhances oxaliplatin efficacy in vivo.
We agree with the reviewer that such experiments would be incredibly interesting to do, however doing animal studies was outside the scope of this study, and given time restraints and resources it would not be realistic to include these studies in the revision of this manuscript.
- Validate Aloperine Specificity: Perform target engagement studies (e.g., CETSA, SPR, or pull-down assays) to confirm direct interaction between aloperine and VPS4A.
We feel we have demonstrated this in section 3.6 (where VPS4 protein expression is inhibited in a dose-dependent fashion by aloperine) to sufficiently justify aloperine’s use as an inhibitor of VPS4 proteins in this study, and we also cited a publication (Guo et al) who have carried out experiments confirming VPS4 specificity. As we have acknowledged its limitations in the discussion section as a VPS4A-specific inhibitor, and that further usage outside of this proof-of-principle demonstration here is not our goal, then we did not think that extensive target engagement studies were warranted in this study.
- SW480 cells: The authors said that among the 10 colon cancer cells used in this study, SW480 colon cancer cells showed a special response to this study, so they performed additional experiments. However, there is no explanation at all as to why SW480 colon cancer cells showed such responses. In general, in studies using cancer cells, it may be due to the characteristics of specific oncogenes, tumor suppressor genes, or other genes, so I hope the authors will discuss this part in the Discussion section.
We have now included a hypothesis of why SW480 and KM12 cells may be particularly useful for these studies in the discussion section.
- English: I feel embarrassed to point out English-related issues in a paper submitted by British confectioners. However, I cannot help but make a few points about English writing. 1) In general, protein or compound names are not necessarily capitalized in the middle of a sentence. In this paper, there are many cases where the first letter of the word for protein or compound is capitalized when it comes to abbreviations, so please correct them all. 2) There are cases where specific cell names are written differently, so please correct them. Examples: HT55 at Line 84 vs. HT-55 at Line 87; KM12 vs. KM-12; etc.
Please see response to point 9
- Abbreviations: The use of abbreviations when writing a paper has many advantages besides simplicity of expression. To use an abbreviation, first write the abbreviation in parentheses after the full name, and then use the abbreviation from Introduction to the final Conclusion. Abbreviations should only be used if they are repeatedly used and if they are not used again, only the full name should be used. Whether it's an abstract or a text, systematically proofread each abbreviation to make sure there are no abbreviations that have been redefined repeatedly.
Please see response to point 9
- In cases where abbreviations are used within figures or tables, please list these abbreviations along with their corresponding full names in the figure legends or at the bottom of corresponding tables. If there are two or more abbreviations, arrange them in alphabetical order.
Please see response to point 9
- Materials and Methods section - When naming a particular chemical company, you must provide location information such as company name, city, and/or state (abbreviation in the USA and Canada) and country. Once you have named a company with the information, you should only mention a company’s name thereafter.
Please see response to point 9
- Notation of units: In scientific papers, almost all units, except for temperature and percentage, are written with a space between the number and the unit. However, in this paper, numbers and units are written together without spaces in both the figures and the text, which is very distracting.
We have extensively reviewed and proof-read the revised manuscript and hopefully addressed any potential issues with the English language as the reviewer has helpfully highlighted in comments 5, 6, 7, 8, and 9.
- Figure 2: Typically, each photo is accompanied by a bar scale; however, in this instance, only the final photo features one.
We have now included error bars on each figure as highlighted by the reviewer
[Minor concerns]
- Figure 1a: ‘KDa’ should be written as ‘kDa’.
Figure has been adjusted accordingly
- Table 3: IC50 should be written as IC50.
Table has been adjusted accordingly
- Line 429: In the case of human genes, they should be written in italics.
This has been adjusted
- References: Adjust the citation style of the references to conform to the cells guidelines, and ensure that any missing volume and page numbers are accurately included. Examples: 1, 2, 3, 5, 6, 7, 22, 32, 33, etc.
Please see response to point 9 above
Round 2
Reviewer 1 Report
Comments and Suggestions for Authors
The authors have addressed all concerns.
Reviewer 3 Report
Comments and Suggestions for Authors
Most of the issues pointed out during the first review process have been appropriately corrected, and the completeness of the paper has also improved. Therefore, I recommend that this paper be accepted.